# Assemblage Structure of Ichthyoplankton Communities in the Southern Adriatic Sea (Eastern Mediterranean)

**DOI:** 10.3390/biology12111449

**Published:** 2023-11-19

**Authors:** Alessandro Bergamasco, Roberta Minutoli, Genuario Belmonte, Daniela Giordano, Letterio Guglielmo, Anna Perdichizzi, Paola Rinelli, Andrea Spinelli, Antonia Granata

**Affiliations:** 1Institute of Marine Sciences, National Research Council (CNR-ISMAR), Arsenale-Tesa 104, Castello 2737/F, 30122 Venezia, Italy; alessandro.bergamasco@ve.ismar.cnr.it; 2Department of Chemical, Biological, Pharmaceutical and Environmental Sciences, University of Messina, Viale Ferdinando Stagno d’Alcontres 31, 98166 Messina, Italy; antonia.granata@unime.it; 3CONISMA LRU Lecce, Department of Biological and Environmental Sciences and Technologies, University of Salento, Campus Ecotekne, 73100 Lecce, Italy; genuario.belmonte@unisalento.it; 4Institute for Marine Biological Resources and Biotechnology (IRBIM), Via S. Raineri 86, 98122 Messina, Italy; daniela.giordano@irbim.cnr.it (D.G.); anna.perdichizzi@irbim.cnr.it (A.P.); paola.rinelli@irbim.cnr.it (P.R.); 5Zoological Station “Anton Dohrn”, Villa Comunale, 80121 Napoli, Italy; letterio.guglielmo@szn.it; 6Research Department, Fundación Oceanogràfic de la Comunitat Valenciana, Oceanogràfic, Carrer d’Eduardo Primo Yúfera 1, 46013 Valencia, Spain; andreaspinelli1990@gmail.com

**Keywords:** ichthyoplankton, community structure, spatial distribution, weighted mean depth (WMD), BIONESS, hydrology, southern Adriatic Sea, Otranto Channel, *Cyclothone braueri*

## Abstract

**Simple Summary:**

This study describes the composition, abundance, spatial distribution and differences in day/night vertical distribution of ichthyoplankton in the southern Adriatic Sea. Samples were collected from 9 to 18 May 2013, in multiple layers from near the seabed to the surface by the electronic multinet EZ-NET BIONESS (Bedford Institute of Oceanography Net Environmental Sampling System). A total of 20 species, belonging to 20 genera and 13 families, were identified. The community was dominated by Gonostomatidae, followed by Engraulidae, Myctophidae and Photychthaidae. The most abundant species was *Cyclothone braueri*, followed by *Engraulis encrasicolus*, *Ceratoscopelus maderensis*, *Cyclothone pygmaea*, *Vinciguerria attenuata* and *Myctophum punctatum*. An inshore/offshore increasing gradient in biodiversity and abundance was observed. Different weighted mean depths (WMD) were observed for larvae and juveniles. No diel vertical migrations were observed. The high abundance of meso-bathypelagic species in the upper 100 m confirms the epipelagic zone as an important environment for the development of the larval stages of these fish.

**Abstract:**

Studies based on fish early life stages can provide information on spawning grounds and nursery areas, helping to determine the implications for stock biomass fluctuations of recruitment variability. This study describes the composition, abundance, spatial distribution and differences in day/night vertical distribution of ichthyoplankton in the southern Adriatic Sea. Samples were collected within the framework of the COCONET project (Towards COast to COast NETworks of marine protected areas) from 9 to 18 May 2013 by the *R/V Urania*, using the electronic multinet EZ-NET BIONESS (Bedford Institute of Oceanography Net Environmental Sampling System). A total of 20 species, belonging to 20 genera and 13 families, were identified. Of the collected larvae, 74.3% were meso- or bathypelagic species, 24.7% were epipelagic and 0.9% were demersal. The community was dominated by Gonostomatidae, followed by Engraulidae, Myctophidae and Photychthaidae. The most abundant species was *Cyclothone braueri* (45.6%), followed by *Engraulis encrasicolus*, *Ceratoscopelus maderensis*, *Cyclothone pygmaea*, *Vinciguerria attenuata* and *Myctophum punctatum*. An inshore/offshore increasing gradient in biodiversity and abundance was observed. Different weighted mean depths (WMDs) were observed for larvae and juveniles. No diel vertical migrations were observed. The high abundance of meso- or bathypelagic species in the upper 100 m confirms the epipelagic zone as an important environment for the development of the larval stages of these fish.

## 1. Introduction

The Mediterranean is known as an oligotrophic sea. A high environmental diversity at both regional and local scales is linked to its varied coastline and bathymetry, further to a high seasonality [1].

Larval fish assemblages belong to the meroplankton community, and their abundance and composition are linked to the reproductive cycles of the adult fish. Marine fish larvae can be classified depending on habitat and the depth of the areas in which the adults live and spawn [2,3,4]. Coastal species are found from the coastline to the edge of the continental shelf (above 200 m depth), and their spawning period is spring/summer. Spawning occurs in superficial layers or in shallow demersal habitats, and larvae produced are carried to deeper oceanic waters [4,5,6,7]. Species inhabiting the pelagic waters can be classified according to the layer in which they live: epipelagic (0–200 m), mesopelagic (200–1000 m) and bathypelagic (1000–4000 m). The spawning period is different for epipelagic and meso- or bathypelagic species; the former spawn between spring and summer, while the latter spawn throughout the year, mainly above 500 m, although some species show one or two spawning peaks [3,4,8].

Studies carried out on fish early life stages can provide information on spawning grounds and nursery areas, which is useful for ascertaining recruitment variability and the resultant stock biomass variability [9,10]. Likewise, studies on the composition, density and spatial distribution of fish larvae in relation to environmental parameters are very important, since the survival and growth of many fish species are linked to the impact of the marine environment on these early developmental stages [11]. The seasonality and the duration of their meroplanktonic life stage lead to different spatial and temporal distributions of fish larvae [12,13,14]. Marine fish larvae can be distributed throughout the whole water column, although many biotic and abiotic factors can influence their occurrence depth [15,16]. The patterns of distribution of the spawning products are initially set by the adults, but different environmental and biological inputs can influence larvae distribution, abundance, growth and survival [16]. The analysis of the spatio-temporal distribution and abundance of ichthyoplankton in relation to oceanographic parameters may provide insights into potential modification of spawning strategies for physical and biological processes [17]. The ability of larvae to remain at a given depth depends not only on their swimming ability, but also on their motivation to stay at that depth. Furthermore, the distribution of fish larvae along the water column is correlated with the presence and position of the thermocline, although some studies give little importance to this physical condition [18,19]. Neilson and Perry (1990) [15] concluded that changes in the depth at which fish larvae are located are probably regulated by endogenous factors but appear to be triggered by environmental factors, such as light levels, prey and predator density, hydrographical conditions, and turbulence. Many micronekton species carry out diel vertical migrations (DVM) from deeper layers to more superficial ones for trophic reasons, feeding in the epipelagic zone at night and returning downwards at sunrise to the meso- or bathypelagic zones [20,21].

This study describes the composition, abundance, spatial distribution and differences in day/night vertical distribution of ichthyoplankton in the southern Adriatic Sea. Potential relations with environmental parameters, chlorophyll *a* concentration and water masses are examined. This ichthyoplankton survey provides useful data for the assessment of relevant parameters for commercially important fish populations (spawning stock biomass, recruitment) and also increases understanding of the agents responsible for structuring larval assemblages, as already shown [14,22].

## 2. Materials and Methods

### 2.1. Study Area

The Adriatic Sea is the most productive region of the Mediterranean Sea. The shallow southern Adriatic is subject to periods of strong wind mixing, even during the summer months of peak spawning [19]. The study area comprises the southern Adriatic basin (maximum depth 1200 m) and the Otranto Channel which connects it with the Ionian Sea, a key region in the oceanography of the eastern Mediterranean basin [23]. The presence of four main water masses has been identified in the southern Adriatic basin [24,25]. The epipelagic layer (0–200 m) is characterized by the southward flow of the Adriatic surface water (ASW) along the Italian side and the Ionian surface water (ISW), saltier (S > 38.25) and warmer (T > 15 °C) than the ASW, entering the Adriatic along the Albanian side of the Otranto Channel. The upper mesopelagic layer (200–600 m) hosts the water formed by its mixing with the intruding Levantine Intermediate Water (MLIW), defined by a saltier core (S > 38.75) in the 200–600 m layer. Underneath, a denser water mass, the Adriatic deep water (ADW; T 13.3 °C; S 38.70) occupies the deeper part of the Adriatic pit and outflows through the Otranto Channel following, approximately, the 900 m isobath [26]. The periodic reversal of the upper circulation in the Ionian Sea (the BIoS mechanism) alternately advects Atlantic (AW) or Levantine (LIW) waters in the southern Adriatic basin and influences the biological processes and biogeochemistry of the region.

### 2.2. Sampling Procedure

Zooplankton samples were collected within the framework of the COCONet project (Towards COast to COast NETworks of marine protected areas) during the first COCONet-WP11 multidisciplinary oceanographic cruise, which took place between 9 and 18 May 2013 in the southern Adriatic Sea aboard the *R/V Urania*. Sampling was carried out in seventeen stations on the Apulian and Albanian coastal and pelagic waters, including the Channel of Otranto (Figure 1).

Sampling details are reported in Table 1. A total of 146 zooplankton samples were collected in different layers from the surface to a few metres above the seabed, along a 0–1100 m water column. Sampling was carried out using the electronic multinet EZ-NET BIONESS (Bedford Institute of Oceanography Net Environmental Sampling System) [27], 0.25 m^2^ mouth, equipped with 10 nets of 230 μm mesh size and a multi-parametric probe SBE 911plus (Sea-Bird Scientific, Bellevue, WA USA) to continuously record temperature, salinity, towing depth and fluorescence. Initial real-time data on depth (m), temperature (°C), salinity and fluorescence (μg L^−1^ Chl *a*) were processed with Ocean Data View (ODV) software 5.5.2 version to obtain a picture of the physicochemical structure of the water column. Flow velocity and filtration efficiency were monitored by two internal and external flowmeters. The BIONESS was deployed at low speed along an oblique path to the maximum selected depth to be investigated, with the first net open to enable filtering of the water during the descent. Then, the collection of useful samples began, towing the BIONESS upwards obliquely at a speed of 1.5–2 m s^−1^, with the second net open, until it closed at the predefined depth and the next net opened. The opening and closing of each net was managed onboard the oceanographic vessel at selected sampling layers according to biological and physical features of the water column observed during the instrument’s descent.

From surface to 100 m depth, the sampled layer thickness was from 10 m to 40 m; from 100 and 1100 m depth the layer thickness was wider, from 50 m to 300 m. All the sampled strata for each station, with an indication of filtered volume, time of sampling, temperature, salinity and fluorescence values are reported in Appendix A. Samples were collected both during daytime (Sts. L41, S3, S7, S8, S16C, S15, S22, S23, S20, S19, S14, S11) and nighttime (Sts. S1, S10, S21, S24, S25) (see Table 1). Local sunrise and sunset times were 05.49 and 20:45 (GMT + 2:00), respectively. The filtered volume of water for each layer ranged between 25 and 108 m^3^, depending on the thickness of the sampled layer. On board, the samples were preserved in 4% buffered formaldehyde seawater solution.

### 2.3. Laboratory Zooplankton Analysis

In the laboratory, subsamples of different volumes, taken from the original sample, were observed under a Leica Wild M10 stereomicroscope. The whole sample was examined for the identification and enumeration of macrozooplankton and micronekton. All the specimens of each taxon were counted and identified at the highest taxonomic level. Zooplankton abundance was calculated by dividing the total number by the filtered water volume and expressed as individuals m^−3^.

Fish larvae and juveniles were sorted out from the entire sample and counted under the Leica Wild M10 stereomicroscope. In this case, specimens were identified to the lowest possible taxonomic level according to the available descriptions. The ichthyoplankton vertical abundance in each sampling station was estimated from the total specimens counted in any sampled layer divided by the volume of the total filtered water and expressed as number of individuals per 100 m^3^. The standing crop in the entire water sample column was expressed as individuals per m^2^. Shrinkage due to formaldehyde preservation (ca. 10%) was not considered.

### 2.4. Identification of Diel Patterns

To identify potential Diel Vertical Migration (DVM), the weighted mean depth (WMD) of fish larvae was estimated for each sampled station, considering the time of sampling, according to the following equation by Barange (1990) and Andersen and Sardou (1992) [28,29]:WMD = Σ(ni × zi × di)/Σ(ni × zi)(1)
where ni is the number of ind. 100 m^−3^ in the i layer, zi is the thickness of the layer, di is the depth of a sample i (mean depth of the sampled stratum, e.g., 90 m for a 100–80 m layer).

### 2.5. Statistical Analysis

To better detail the presence of different water masses in the area, a section across the NW–SE direction in the middle of the region was defined (Sts. S3, L41, S11, S15, S21, S23). To highlight the spatial differences in the water column’s vertical structure in the region, the environmental measurements were pooled according to depth in the epipelagic and upper mesopelagic layers (five 40 m thick layers and eight 50 m thick layers, respectively) and considered as a unique group below 600 m. They were then categorized according to their geographic position in three classes: Italian side (Sts. S1, L41), Pelagic (Sts. S3, S10, S11, S14, S15, S19, S20, S25) and Albanian side (Sts. S7, S8, S16c, S21, S22, S23, S24).

Regarding the sampling strategy, larvae abundance and diversity in relation to relevant factors (factor “layer”: eight levels; factor “water masses”: four levels; factor “geographical position of the stations”: five levels) were studied using R version 4.0.3. to highlight significant differences between and within groups of data (ANOVA and Kruskal–Wallis non-parametric tests on medians in the case of non-homogeneity of variances, coupled with a post hoc Dunn test). The non-parametric Spearman rank correlation test was employed to evaluate the degree of association between two variables. Indicator species analysis (IndVal) was applied to identify species that were representative of different sample aggregations (layers, water masses, geographical area) by using the function “multipatt” of the R package “indicspecies”. Furthermore, distance-based redundancy analysis (dbRDA) was carried out to evaluate similarity patterns in larvae communities related to environmental parameters (longitude, depth, temperature, salinity and chlorophyll *a*). Bray-Curtis distance to non-empty samples was applied. To describe diversity, standard indexes were used (species richness (γ), i.e., number of species in a sample or in a group of samples; alpha diversity (α), i.e., average number of species in a sample and Margalef’s index, i.e., (S − 1)/ln N, where S and N are “total number of species” and “total number of specimens” in a sample, respectively). To give an insight into the change in species composition from local to regional scales (here denoted by “sample” and “water mass”, respectively), Whittaker’s species turnover β-1 (where β = γ/α) was calculated. All the diversity-related indices were calculated using the “vegan” package in R.

## 3. Results

### 3.1. Oceanographic Conditions

The oceanographic state during this cruise has already been described with reference to the θ-S diagram of the collected multiparametric probe profiles [30]. With reference to the section in the NW–SE direction within the area (Figure 2,) the analysis clearly shows the Levantine waters entering the Otranto Channel from south. These waters are characterized by higher salinity with an intruding core of S > 38.85 at around 400 m depth at the entrance. They directly influence the salinity in the upper mesopelagic and epipelagic layers by mixing with the ISW up to the southern Adriatic pit (St. S15) and, through the counterclockwise circulation of the ISW, also up to the outer northern part of the Italian shelf (St. S3).

In the lower mesopelagic layer (>600 m, Sts. S15, S11) and at the shelf-break (>450 m, St. L41), the presence of a colder and less-salty water mass characterized by a density anomaly >29.2 can be observed. This is the ADW that originated in the southern basin (SAdDW) or formed in the northern basin (NAdDW) and that will outflow southward from the Otranto channel close to the seabed.

Temperature and salinity measurements collected during the BIONESS tows throughout the water column are shown in Figure 3.

The thermocline is already well developed in S14 and S19 (up to 3 °C), and in general the mixed-layer depth can be placed between 20 m and 30 m. The Italian coastal environment was affected by colder and less-salty waters than the Albanian side throughout the whole water column. The structure of the water column shows the presence of the ASW and the ISW in the epipelagic layer. In particular, the Italian shelf up to 120 m is influenced by the fresher input of surface water from the north Adriatic (ASW), while the ISW dominates the rest of the epipelagic layer. Due to the strong effect of the LIW in the southern Adriatic in this period, in the upper mesopelagic layer salinity approaches 38.9 on the Albanian side at 350–400 m, where the difference with the Italian side is quite evident (DS ≅ 2).

### 3.2. Zooplankton

Mean total zooplankton abundance was 442 ± SD 258 ind. m^−3^. Copepods dominated, representing from 72 to 91% of the total zooplankton, with a mean abundance of 401 ± SD 236 ind. m^−3^. Zooplankton abundance was higher on the Italian than the Albanian coast (408 ± SD 812 ind. m^−3^ and 219 ± SD 53 ind. m^−3^, respectively). Spring holoplankton accounted for the main part of the zooplankton (85–98%). Abundance peaks of the most representative species occurred at the chlorophyll *a* maximum depth, namely between 20 and 40 m depth and between 60 and 80 m depth.

### 3.3. Fish Larvae

#### 3.3.1. Composition

The larval fish community consisted of a total of 1136 individuals represented by larvae, post-larvae and juveniles (these last represented only by 172 specimens, 15.1% of the total catch). Mean total abundance was 102.61 ± SD 10.17 ind. m^−2^. A total of 20 species were identified (plus five genera of unidentified species), belonging to 20 genera and 13 families (Table 2). Of the larvae identified, 74.3% were meso-bathypelagic fish species, 24.7% were epipelagic and only 0.9% were demersal. The community was dominated by Gonostomatidae (595 specimens, 52.4% relative abundance, 52.05 ind. m^−2^ mean abundance), followed by Engraulidae (268 specimens, 23.6% relative abundance, 27.58 ind. m^−2^ mean abundance), Myctophidae (201 specimens, 17.7% relative abundance, 16.82 ind. m^−2^ mean abundance) and Photychthaidae (45 specimens, 4% relative abundance, 3.87 ind. m^−2^ mean abundance).

Together, these four families accounted for more than 97%, while each of the other nine families identified accounted for less than 1%. There were larvae of fish that inhabit different depths/realms as adults: larvae of epipelagic species, e.g., carangid *Trachurus trachurus*, engraulid *Engraulis encrasicolus* and scombrid *Thunnus thynnus*; of meso- and bathypelagic species, e.g., gonostomatid *Cyclotone braueri* and *C. pigmaea*, myctophid *Cerastoscopelus maderensis*, *Myctophum punctatum*, *Benthosema glaciale* and *Hygophum benoiti*, photychtaid *Vinciguerria attenuata*, sternoptychid *Argyropelecus hemigymnus* and stomiid *Chauliodus sloani*; and of demersal species, e.g., labrid *Coris julis*.

*Cyclothone braueri* was the most abundant and frequent species in the samples, being dominant in the whole study area, collected with the highest frequency (94.1% of the 17 sampled stations), with a total of 518 collected specimens, 45.6% of relative abundance and 45.41 ind. m^−2^ of mean abundance (Table 2). The second most abundant species was *E. encrasicolus*, followed by *C. maderensis*, *C. pygmaea*, *V. attenuata* and *M. punctatum*. Each of the other identified species represented less than 1%.

Myctophidae was the family represented by the largest number of species (seven) plus two genera of unidentified species (*Hygophum* sp. and *Lobianchia* sp.), with *C. maderensis* dominating as regards mean abundance and percentage relative abundance (Table 2). The Gonostomatidae family was represented by two species (*C. pygmaea* and *C. braueri*), as was the Sgombridae family (*Auxis rochei* and *T. thynnus*). Each of the other 10 families was represented by only one species, for example Engraulidae, which was represented by *E. encrasicolus*.

#### 3.3.2. Horizontal Distribution

Among the coastal stations (S1, S3, S7, S25, S19), St. S7 showed the highest abundance value (128.25 ind. 100 m^−3^) and St. S1 the lowest (16.7 ind. 100 m^−3^). Among the offshore pelagic stations (L41, S10, S8, S16C, S15, S22, S21, S23, S24, S20, S14, S11), the highest abundance was found in St. S24 (200.99 ind. 100 m^−3^) and the lowest in St. S15 (28.03 ind. 100 m^−3^).

The horizontal distribution of the six most abundant species is shown in Figure 4. *C. braueri* was present in all the sampling stations, except the Italian coastal St. S1, with maximum abundance in St. S24 (287.29 ind. m^−2^), followed by St. S20 (186.05 ind. m^−2^), St. S23 (99.61 ind. m^−2^), St. S21 (44.3 ind. m^−2^) and St. S25 (40.62 ind. m^−2^). The engraulid *E. encrasicolus* was present in eleven stations. Its highest value was observed in St. S22 (259.21 ind. m^−2^), followed by St. S10 (61.75 ind. m^−2^) and St. S24 (56.01 ind. m^−2^). In the other sites, this species showed lower abundance values. *C. maderensis* was present in twelve stations, with the highest abundance being found in St. S24 (99.38 ind. m^−2^), followed by St. S23 (25.36 ind. m^−2^) and St. S10 (17.93 ind. m^−2^). *C pygmaea*, found in eleven stations, was present with the highest value in St. S24 (61.43 ind. m^−2^), followed by St. S10 (9.96 ind. m^−2^) and St. S23 (9.06 ind. m^−2^). *V. attenuata*, found in five stations, showed the highest abundance in St. S21 (11.81 ind. m^−2^), followed by St. S7 (7.97 ind. m^−2^). *M. punctatum*, found in six stations, was most abundant in St. S21 (8.86 ind. m^−2^), followed by St. S8 (6.09 ind. m^−2^).

Applying the Pearson correlation to study the potential relation between abundance and geographical position, a significant positive correlation of 0.22 (t = 2.2119, df = 92, *p*-value = 0.02945) (*p* < 0.05) between fish larvae abundance and longitude was found, meaning that abundances increased eastward going from Italy to the Albanian coast. Conversely, a coefficient of −0.21 (t = −2.0855, df = 92, *p*-value = 0.0398) was found for the correlation between fish larvae abundance and latitude, with abundances that increased going southward to lower latitudes.

No correlation was found between the abundance of fish larvae and either the total abundance of zooplankton or the abundance of copepods.

#### 3.3.3. Vertical Distribution

As regards the vertical distribution, a decrease in fish larvae abundance with increasing depth was observed throughout the study area. In both coastal and pelagic stations, larval fish assemblages occupied the whole of the water column, but the highest values were recorded in the first 100 m. The highest abundances occurred in the 0–40 m layer, in both inshore and offshore sampling stations. Going from 40 m to the maximum sampled depth, an abundance decrease was observed in all the stations, except for St. S11, which showed similar values between more superficial and deeper strata. Sts. L41, S14 and S11 did not show fish larvae in the 0–40 m layer.

Regarding the diel patterns of distribution along the water column, the weighted mean depth (WMD) values (m) for the four most abundant species, considering all the positive (non-empty) sampled layers in all the stations, are shown in Figure 5, separately for larvae and juvenile specimens. *C. braueri* larvae occupied more superficial layers relative to juveniles across all of the sampling times. WMD ranged between 16 m and 90 m depth, with only one station in which larvae showed a deeper WMD, namely 141 m (St. L41). Juvenile specimens showed, instead, a deeper occurrence along the water column, with WMD ranging between 364 m and 757 m. *C. braueri* did not show diel vertical migration (DVM), neither as larvae nor as juveniles.

In addition, *E. encrasicolus* did not show variation in depth occurrence between day- and nighttime. The WMD of larvae ranged between 23 m and 121 m, with one deeper WMD of 300 m (St. S21). Juvenile developmental stages showed deeper WMD of from 300 m to 486 m. *C. maderensis* larvae, similarly to the larvae of other species, showed more superficial distribution relative to juveniles. Larval WMD ranged between 7 m and 90 m. *C. maderensis* larvae did not show DVM. Juvenile WMD are not shown in Figure 5, because only three specimens were found in St. S22, in the layer 400–600 m. *C. pygmaea* as larvae showed WMD of between 5 m and 54 m, with one deeper value of 150 m (St. S10). Juveniles of this species showed deeper WMD, from 300 to 850 m, and without differences between day- and nighttime, similar to the other three species.

Profiles of the vertical distributions for the four most abundant species are shown in Figure 6 and Figure 7, separately for larvae and juvenile specimens, together with chlorophyll, temperature and salinity values. Three pelagic stations in the Otranto Channel (S20, S24, S22), a coastal station near Italy (S3), a coastal station near Albania (S7) and a pelagic central station (S10) were the stations selected.

The vertical distribution of these species in the selected stations confirms a bimodal distribution pattern with a more superficial occurrence for larvae and post larvae and a deeper one for juvenile specimens.

*C. braueri* larvae in St. S20 were concentrated from the surface to 40 m depth, with very low densities up to 200 m depth. Conversely, the juveniles mainly occupied the 300–400 m layer and exhibited scarce presence until 600 m depth. Likewise, in St. S24, the larvae were concentrated in the upper 40 m, with only a few specimens until 100 m, while juveniles occupied a deeper layer, from 400 m to 800 m. St. S22 again showed the same bimodal distribution for larvae and juveniles, mainly in the ranges 0 m to 40 m and 300 m to 600 m, respectively. The two coastal Sts., S3 and S7, did not show any juveniles of mesopelagic *C. braueri*, at bottom depths of 178 m and 190 m respectively, whereas larvae occupied the 20–60 m and 0–40 m layers, respectively. In St. S10, larvae were present in the 80–100 m layer and juveniles from 200 m to 1100 m depth.

Regarding the epipelagic anchovy *E. encrasicolus*, the species showed a deeper occurrence of juveniles relative to larvae. In St. S20, only larvae were found, between 40 m and 80 m. In St. S24, larvae were present between 0 m and 40 m, with a few specimens in the 80–100 m and 200–400 m layers. Juveniles were distributed from 200 m to 800 m depth. In St. S22, larvae were concentrated between 0 m and 60 m. In Sts. S3 and S7, juveniles of *E. encrasicolus* were not found. Larvae were distributed from 20 m to 100 m, with the highest values between 40 m and 60 m in both stations. In the pelagic St. S10, the anchovy larvae were present from 60 m to 200 m and from 400 m to 600 m. The juvenile population was concentrated in the 400–600 m layer.

The third most abundant species, *C. maderensis*, did not show both developmental stages in all the selected stations. In Sts. S20 and S24, only larvae of this species were present, distributed from 20 m to 60 m depth and from the surface to 80 m depth, respectively. Conversely, in St. S22 only juveniles were present, concentrated in the 400–600 m layer. Similarly, in Sts. S3 and S7, only larvae were found, from 20 m to 100 m and from the surface to 60 m, respectively. In St. S10, larvae and juveniles were both observed both in 0–40 m layer.

For *C. pygmaea,* opposite profiles throughout the water column were also observed for larvae and juveniles, albeit with scarce presences and low densities. In St. S20, larvae occupied the 0–40 m layer while juveniles occupied the 300–400 m layer. Also, in St. S24 larvae were concentrated in the upper 40 m, with a few other individuals until 80 m depth. The juveniles were present between 400 m and 800 m, with a few specimens also between 120 m and 200 m. In St. S22, only larvae were found, from the surface to 40 m depth. Similar to other species, St. S3 and S7 did not show juveniles, but only larvae, from 20 m to 40 m and from the surface until 20 m depth, respectively. In the pelagic St. S10, larvae and juveniles were found, in the 100–200 m and 200–400 m layers, respectively.

#### 3.3.4. Fish Larvae Abundance and Biodiversity Relationship with Water Masses

Evaluating the Pearson correlation, no significant correlations were found between abundance and several environmental parameters, like salinity (−0.1), chlorophyll (−0.2) and oxygen (−0.02). A very significant correlation coefficient of 0.54 was found only with the temperature (t = 6.1622, df = 92, *p*-value = 1.873 × 10^−8^) (*p* < 0.001). In fact, the abundance of fish larvae increased where the temperatures were higher.

The analysis of the spatial distribution of fish larvae abundances suggests that differences between water masses were not significant (ANOVA, *p* > 0.05), while a weak significance could be observed between stations when grouped by region according to the geographical proximity to the Italian or Albanian sides (ANOVA, *p* = 0.07) (Figure 8).

As for the relations throughout the water column, the correlation analyses showed that abundance decreased with increasing depth (Spearman’s rank correlation r = −0.28, *p* = 0.005, N = 94). Going into a bit more detail on this, the medians of abundances in the different layers (Figure 9) exhibited a strong significant difference (ANOVA, Kruskal–Wallis, chi-squared = 32.8, df = 8, *p* < 0.001). In particular, the post hoc Dunn test highlighted that abundances in the surface layer (0–40 m) were greater than the rest of the epipelagic layer (60–80 m layer, *p* < 0.05; 80–100 m layer, *p* < 0.05; 100–200 m layer, *p* < 0.001) and the deepest waters (600–800 m layer and >800 m layer, *p* = 0.001), but not significantly greater than the upper mesopelagic layer, which hosts the MLIW (200–600 m layers). IndVal analysis performed on the larval assemblages did not indicate any significant characteristic species for the aggregation by water mass and by geographical position. As for the vertical structure, represented by the aggregation by layer, the species *C. maderensis* was a clear indicator of the 0–60 m layer (A = 0.86, B = 0.46, *p* < 0.05), and *Hygophum* sp. was a clear indicator of the 0–80 m layer, with very high specificity but quite low occurrence (A = 0.97, B = 0.38, *p* = 0.02). Interestingly, the species *Paralepis speciosa*, exclusive to the ADW water mass within this study, was highlighted as a significant indicator of the >800 layer (A = 1, B = 0.33, *p* = 0.04).

Figure 10 shows the constrained classification produced by db-RDA analysis. The first two axes (both weakly significant, *p* < 0.05, ANOVA by axis) explain only 5.4% of the fitted model, with temperature significantly influencing the larvae assemblage (*p* < 0.01, ANOVA by term). Four of the most abundant larvae species were gathered towards the increase of temperature in the halfplane related to the epipelagic layer (ISW and ASW), while *E. encrasicolus* showed greater affinity with increasing longitudes (e.g., the Albanian side) and the upper mesopelagic waters (MLIW).

To compare the diversity in the four water masses, the reciprocal position of the k-dominance curves can be examined (Figure 11). Though they each exhibit a quite different overall species richness, the four curves strictly track each other for the first (3–4) ranked species, and then the more speciose ISW exhibits the flattest slope (beyond the 8th rank). This is related to good levels of completeness for the samples describing this water and could also depend on the relevant import of species from a larger metacommunity, driven by the mixing with MLIW.

Diversity descriptors are summarized in Table 3. Based on the Margalef index, diversity did not significantly vary across the water masses (Kruskal–Wallis test: *p* > 0.05) and was slightly higher in the ISW. Alpha diversity was around two species/sample throughout the region and is surprisingly higher in the ASW, which suggests the need to put more effort into sampling this water mass to better capture the nature of its diversity.

This beta diversity (e.g., the differentiation among samples) is at a maximum for the ISW group—which presumably exhibits a great variability of habitat/niches—but is also relatively high for MLIW.

## 4. Discussion

The interaction between fish larval stages and physical, chemical and biological environmental features plays a pivotal role in recruitment variability. Environmental factors, in fact, strongly influence the distribution of ichthyoplankton, which are an important component of marine plankton communities in the oceanic ecosystem, playing a key role in the marine food webs through bottom-up control [31]. It is important to understand how connectivity can influence survival during the early life developmental stages [4]. The adult fish populations are connected to the survival and successful recruitment through time of the early life stages (eggs, larvae and juveniles) [32]. Larval fish stage distribution and abundance depend upon the adult population size, spawning time and area, and suitable habitat availability [33,34]. It is already known that the spawning period for some fish larval species precedes plankton blooms [35]. Fish larvae, due to weak swimming ability, are passively moved long distances horizontally, perhaps even for hundreds of kilometres [4,34], as a result of various environmental conditions. The space–time variability of oceanographic structures can influence the physiology and behaviour of ichthyoplankton, leading to variations in recruitment and survival [36]. The study of the relation between larvae abundance and environmental conditions gives the opportunity to identify suitable habitats and observe habitat shifts linked to anthropogenic activity that may influence the survival and recruitment of fish larvae [37,38,39]. The pre-flexion stage larvae use their ability to migrate vertically in order to either stay in an area or distribute themselves horizontally [40,41]. Several studies have pointed out that even at the very early stages, larvae migrate vertically, following the general plankton migrations, in a trade-off between finding food and hiding from predators [42]. This migratory pattern is commonly seen in many studies from pelagic locations [43]. In the framework of ichthyoplankton ecology research, this current study was developed to enhance the knowledge of these aspects as they relate to the southern Adriatic Sea, analyzing abundance, community structure, spatial distribution and differences in day/night vertical distribution.

In the area investigated in this study, the most abundant species was the mesopelagic gonostomatid *Cyclothone braueri* (45.6%), which was followed by the epipelagic engraulid *Engraulis encrasicolus* (23.6%) and four other mesopelagic species, namely *Ceratoscopelus maderensis* (10.1%), *Cyclothone pygmaea* (6.2%), *Vinciguerria attenuata* (2.9%) and *Mycthophum punctatum* (1.6%).

*C. braueri*, which represented approximately half of the catches, is already known as the most common mesopelagic fish in the offshore areas of the Mediterranean Sea [44,45,46] and is found in productive shelf areas, where it is associated with coastal upwelling, lower water temperature and higher Chl *a* concentrations [14,47,48,49,50]. *C. braueri* was shown in this study to be associated with the presence of larvae both inshore and offshore as well as an absence of juveniles in coastal stations, a phenomenon likely to be linked to bottom depth. *C. braueri* juveniles did not demonstrate diel vertical migrations, occupying the same layers during both daytime and nighttime. Juveniles of this species are known to occupy the same mesopelagic layer during the whole 24-hour cycle, unlike, e.g., the migratory lanternfishes that engage in DVM [50]. This observation confirms that *C. braueri* is a species that does not migrate along the water column; that has a mesopelagic vertical distribution pattern, as already reported [20,21,50,51]; that lives between 300 and 900 m [20,51] and that shows the highest abundance in the Mediterranean continental slope 400–600 m deep scattering layer (DSL) [44,52]. The larval stage of this species in our study was also shown to be nonmigratory, with the bulk of the population concentrated in the superficial 0–40 m layer during all the sampling times, certainly for trophic reasons. The literature reports that *C. braueri* carries out ontogenetic migrations, which are characterized by eggs and larvae in the euphotic zone [14,50,53,54] and juveniles that move downward to deeper strata during their development.

*E. encrasicolus*, the second most abundant species in the investigated area, represented the most abundant epipelagic fish species both inshore and offshore relative to some others that were present but had very few specimens (*Trachurus trachurus*, *Paralepis speciosa*, *Chromis chromis*, *Auxis rochei*, *Thunnus thynnus*). *E. encrasicolus*, a common species in the Mediterranean Sea, did not show DVM and showed a bimodal pattern of vertical distribution, like *C. braueri*. Linked with food availability, the bulk of larvae were concentrated in the first 100 m, whereas the juvenile developmental stages were more abundant between 400 and 500 m, a shallower depth compared with *C. braueri*. It was already known, in fact, that mesopelagic fish larvae had a deeper occurrence in the water column than pelagic fish larvae [50].

The species that came next in order of decreasing abundance in the investigated area (*C. maderensis*, *C. pygmaea*, *V. attenuata*, *M. punctatum*) are all meso-bathypelagic species. Considering the total caught, in fact, the mesopelagic species were much more abundant (74.3%) compared with the epipelagic species. The high abundance of mesopelagic fish species larvae found in coastal areas of the southern Adriatic Sea has already been reported for other areas and linked to oceanic water intrusion towards the coast [55,56,57,58]. Mesopelagic fish have an important role in the oceans, representing prey for other species of commercial value and so being a fundamental link in the food chain [50]. In the coastal stations, species diversity and abundance were lower with respect to the pelagic species. In the inshore stations, juvenile stages of mesopelagic species were not observed (e.g., see Figure 4, Sts. 3 and 7), owing to the shallow depth. In summary, the study area—the southern Adriatic Sea—is therefore characterized by a typical shelf-dwelling ichthyoplankton community formed by developmental stages of fish that live in different habitats when adults, and an offshore-dwelling ichthyoplankton community, formed by pelagic, meso- and bathypelagic larvae and juveniles.

An increasing gradient in ichthyoplankton abundance was highlighted going from the Italian to the Albanian coast, relying on the presence of colder and less-salty waters along the Italian coast than on the Albanian side throughout the whole water column. In fact, a positive correlation between ichthyoplankton abundance and temperature was found. Of less importance seems to be food availability, which appeared not to be a limiting factor since no correlations between total zooplankton and fish larvae abundances and between copepod and fish larvae abundance were found.

Both the most abundant mesopelagic species and anchovy showed their highest densities in the Otranto Channel and in the southernmost pelagic stations, always on a bottom depth of greater than 700 m. The horizontal distribution pattern found in our samples for larval stages is in agreement with previous findings in other areas [59] and in the same area [47]. Results confirmed by statistical Pearson correlation underlined increasing fish larvae abundance going southward. Our study site is located in a region where the relatively eutrophic waters from the Adriatic Sea mix with the oligotrophic waters of the Ionian Sea. The sampling was carried out in a season in which many coastal fish, which spawn in spring, reach their maximum abundance and the water column is still at the beginning of the surface-warming phase [47]. In the statistical analyses, in fact, no correlation between fish larvae abundance and water masses was found. This mixing leads to zooplankton aggregation [60] that represents a food source for adult fish [47,61,62] and allows fish larvae to grow rapidly throughout the early summer [12,55]. That the neritic environment is suitable for fish larvae ontogenesis is something that is already known [47].

As for vertical distribution of total ichthyoplankton, both abundance and biodiversity decreased going from the surface to deeper strata, with maximum abundance in the upper 100 m and highest values in the 0–40 m layer. This can be confirmed by the positive correlation between fish larvae abundance and temperature, explaining the highest abundance in the superficial warmest waters. Different patterns of vertical distribution for larvae and juveniles were highlighted for the most abundant species, both meso-bathypelagic and epipelagic ones. Larvae showed a more superficial distribution linked to trophic reasons compared with later developmental stages, which had deeper ranges of occurrence, similar to the adult stage. Even though the bulk of the ichthyoplankton community occupied the first 100 m, the community inhabited the entire water column, with a few specimens even reaching 1000 m depth, maybe because of quite homogeneous temperatures until the maximum sampled depth.

## 5. Conclusions

Ichthyoplankton surveys can provide useful data for the assessment of relevant parameters for commercially important fish populations (spawning stock biomass, recruitment) and can also improve available knowledge on the agents that structure larval assemblages. The study of mesopelagic fish and their developmental stages is fundamental to understanding food web functioning in a marine environment where bi-directional mechanisms of vertical coupling throughout the whole water column are of paramount importance. For example, the most abundant mesopelagic species observed in this study, *C. braueri*, represents a key species in the energy transfer from zooplankton to higher trophic levels, such as myctophids and larger epipelagic fish [50,63].

This study has shown a peculiar geographical distribution of fish larvae in the southern Adriatic Sea, with abundance values increasing from the Italian coast towards the Albanian coast and from north to south of the region. A vertical gradient was highlighted throughout the water column, with decreasing abundance going from the surface to deeper layers, and with maximum values in the upper 100 m of both inshore and offshore areas, in accordance with the developmental stage of the specimens. Larval stages were shown to occupy superficial layers during the whole day due to their feeding behaviour, while for juveniles a deeper occurrence was shown. The high abundance of meso-bathypelagic larval developmental stages in the upper 100 m of the water column highlighted in this study confirms the epipelagic zone as a fundamental area for the development of early phases of these fish [47,64,65].

## Figures and Tables

**Figure 1 biology-12-01449-f001:**
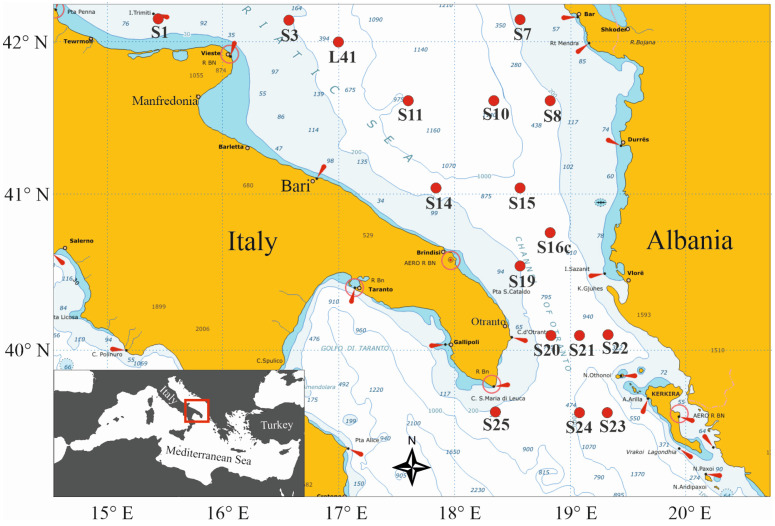
Map showing the stations in the south Adriatic Sea that were sampled from 9 to 18 May 2013 during the *R/V Urania* COCONet-WP11 cruise.

**Figure 2 biology-12-01449-f002:**
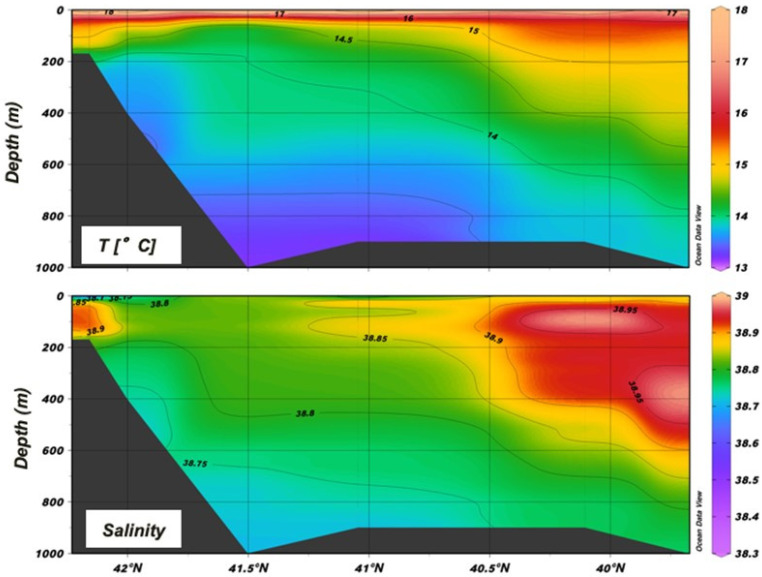
Section across the NW–SE direction in the middle of the region (stations S3, L41, S11, S15, S21 and S23 were considered).

**Figure 3 biology-12-01449-f003:**
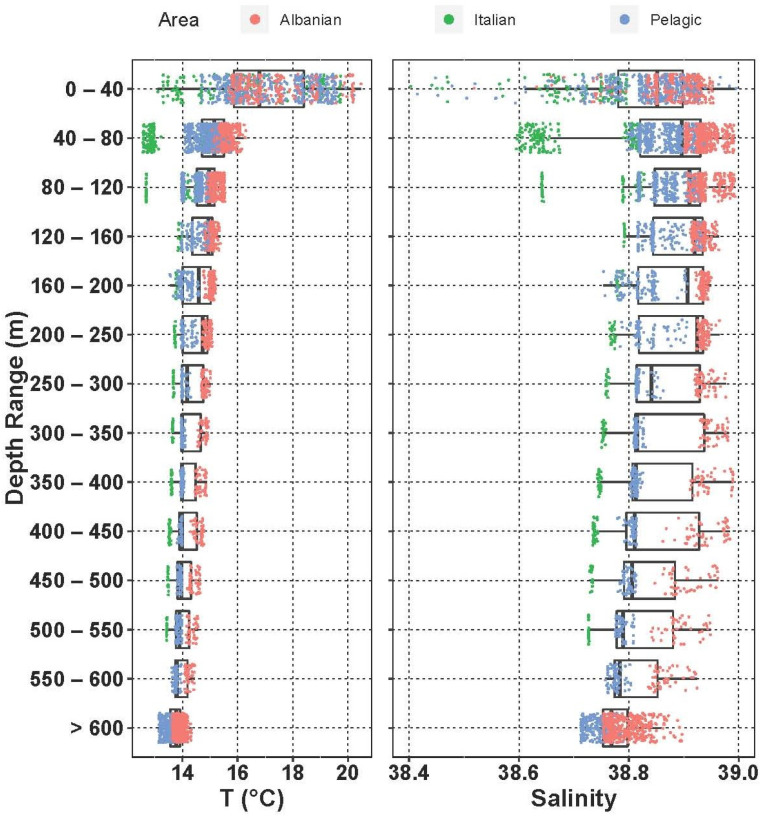
Vertical structure of the water column in the different areas of the region.

**Figure 4 biology-12-01449-f004:**
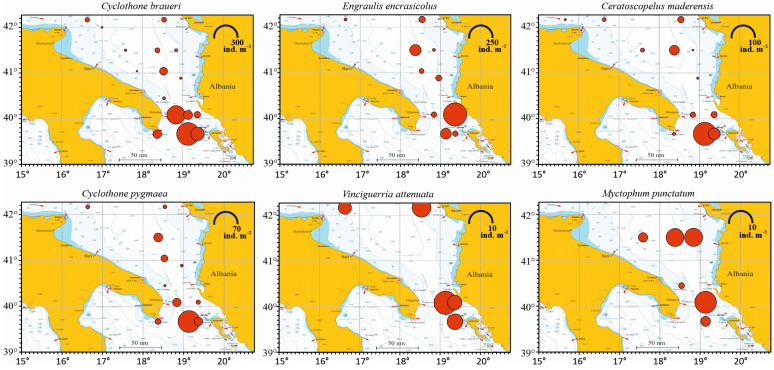
Abundance representation, expressed as ind. m^−2^, of the six most abundant species of fish larvae in the south Adriatic Sea. Values for each station come from the integration of all the sampled water column abundances expressed on a 1 m^2^ basis. Note differences in abundance scale.

**Figure 5 biology-12-01449-f005:**
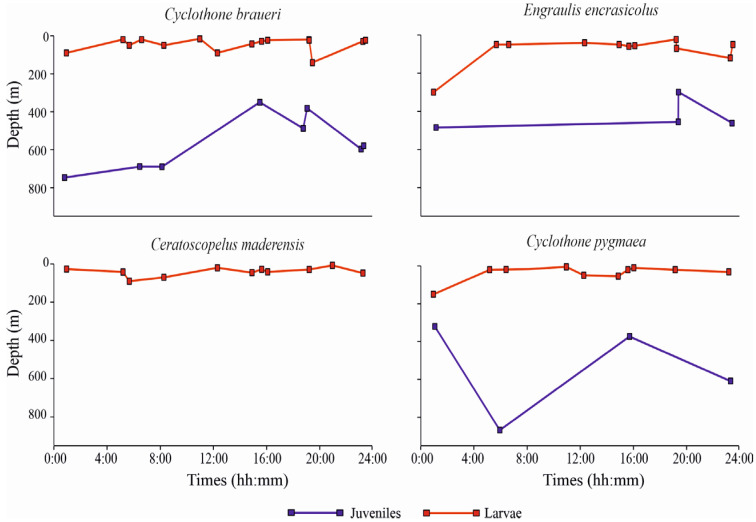
Weighted mean depths (WMD) for the four most abundant species of fish larvae (separately for larvae and juveniles).

**Figure 6 biology-12-01449-f006:**
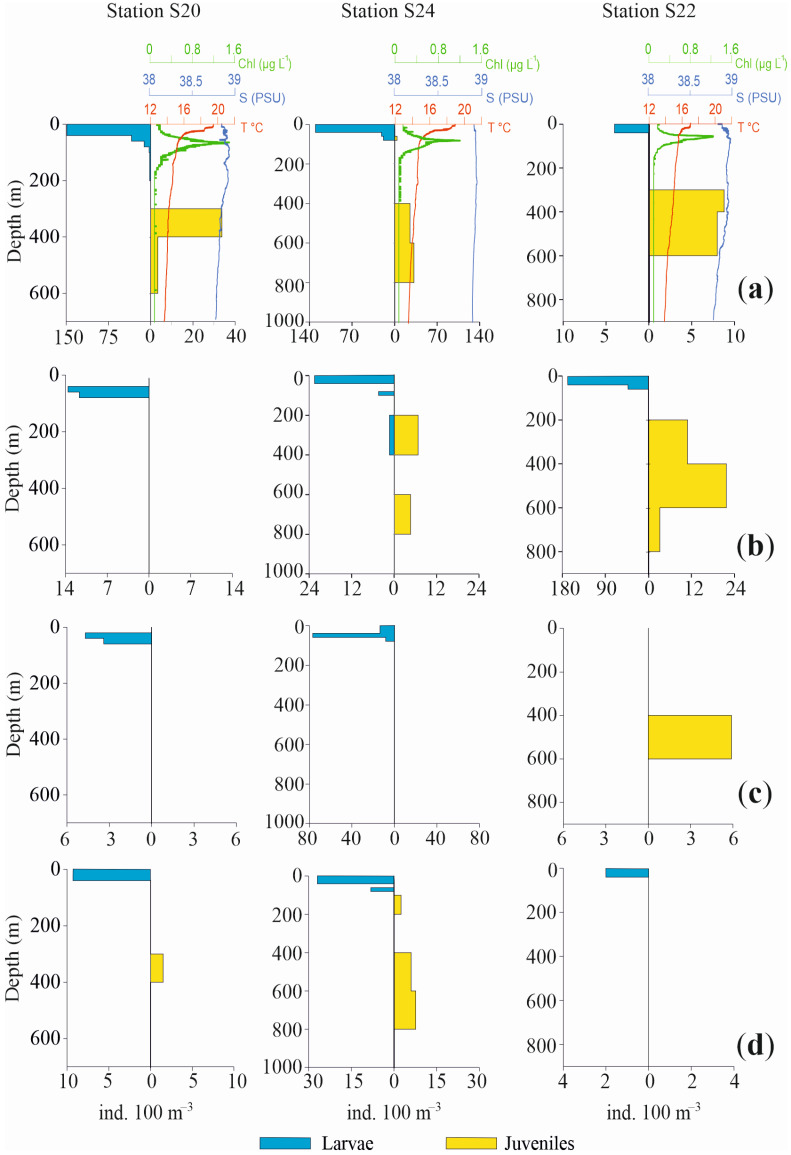
Profiles of abundance vertical distribution for the four most abundant species of fish larvae in Sts. S20, S24, S22, together with chlorophyll, temperature and salinity values ((**a**): *Cyclothone braueri*, (**b**): *Engraulis encrasicolus*, (**c**): *Ceratoscopelus maderensis* and (**d**)*: Cyclothone pygmaea*). Profiles are separately shown for larvae and juveniles. Note differences in abundance and depth scales.

**Figure 7 biology-12-01449-f007:**
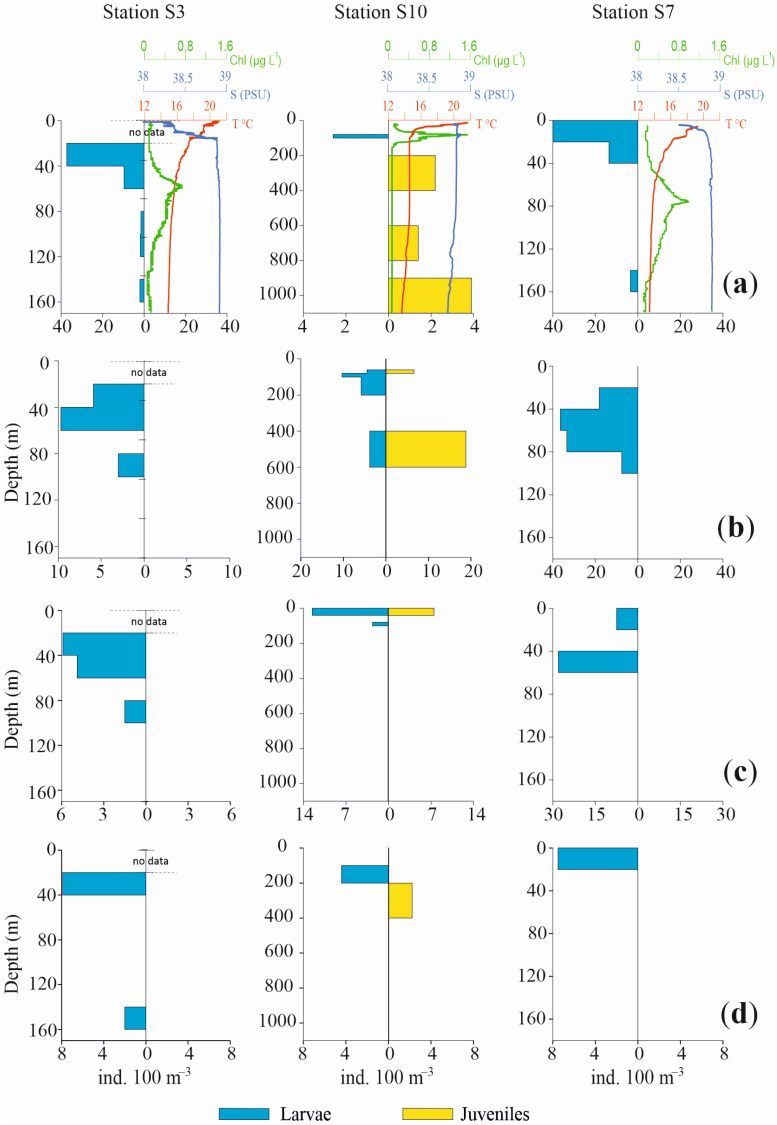
Profiles of abundance vertical distribution for the four most abundant species of fish larvae in Sts. S3, S10, S7, together with chlorophyll, temperature and salinity values ((**a**): *Cyclothone braueri*, (**b**): *Engraulis encrasicolus*, (**c**): *Ceratoscopelus maderensis* and (**d**): *Cyclothone pygmaea*). Profiles are separately shown for larvae and juveniles. Note differences in abundance and depth scales.

**Figure 8 biology-12-01449-f008:**
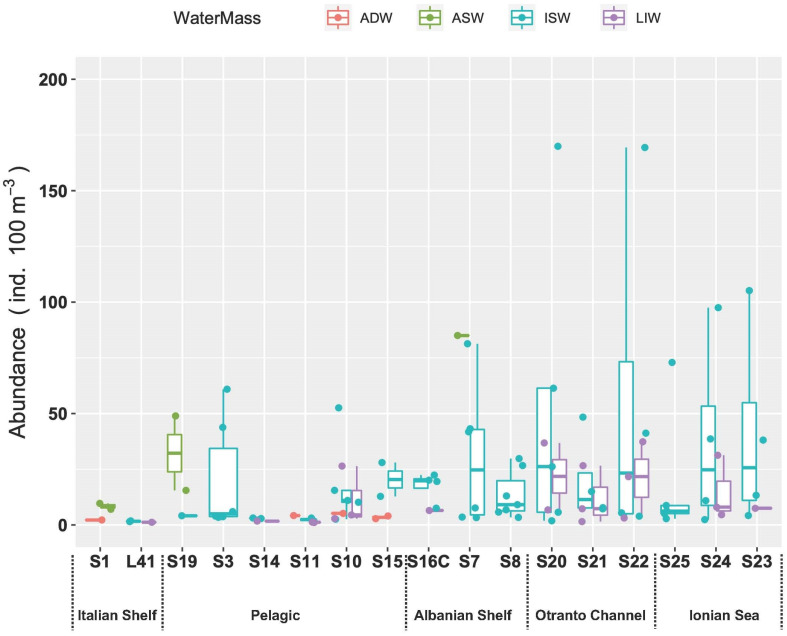
Fish larvae abundance in the study area assigned to the different water masses. Stations are ordered in NW–SE direction.

**Figure 9 biology-12-01449-f009:**
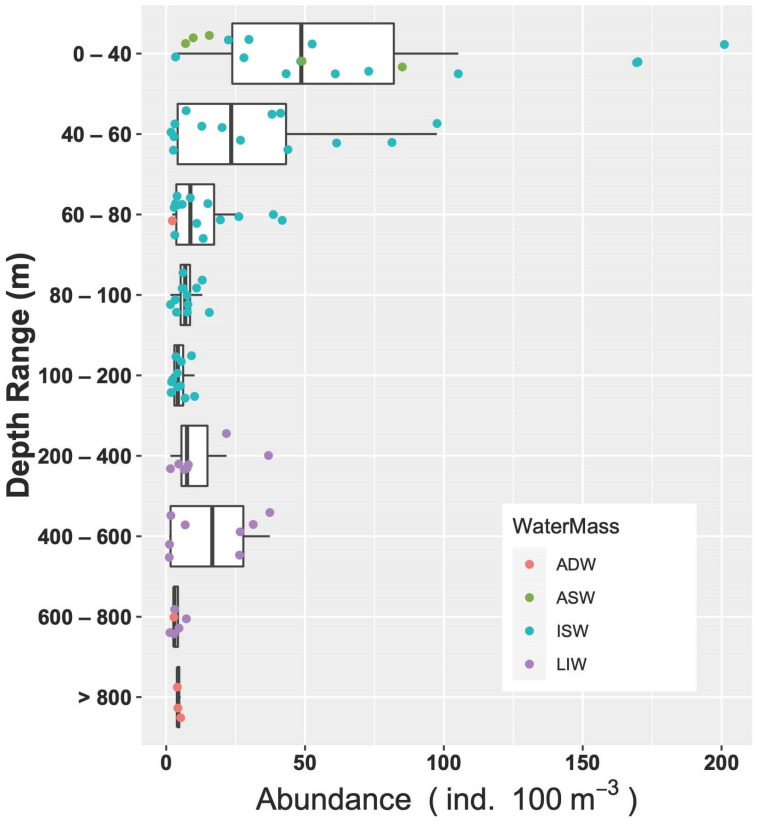
Vertical distribution of larvae abundances assigned to the different water masses.

**Figure 10 biology-12-01449-f010:**
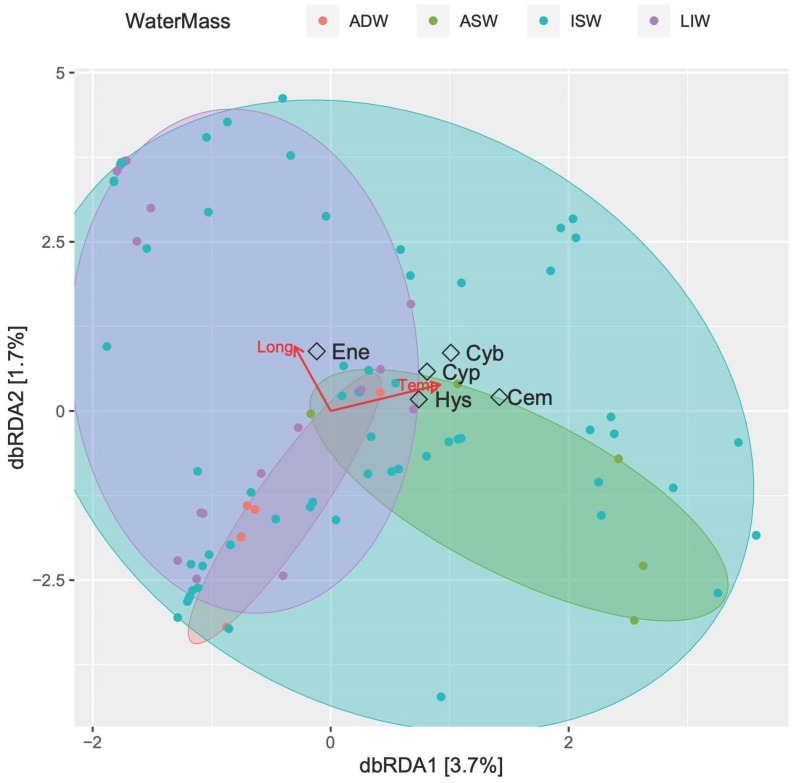
Constrained classifications by distance-based redundancy analysis (db-RDA) ordination diagram of fish larvae species in southern Adriatic Sea. Red arrows indicate significant explanatory variables (longitude and temperature). The samples are labelled according to their water-mass cluster membership. The five most abundant species are shown (Cyb: *C. braueri*, Cyp: *C. pygmaea*, Cem: *C. maderensis*; Hys: *Hygophum* sp.; Ene: *E. encrasicolus*).

**Figure 11 biology-12-01449-f011:**
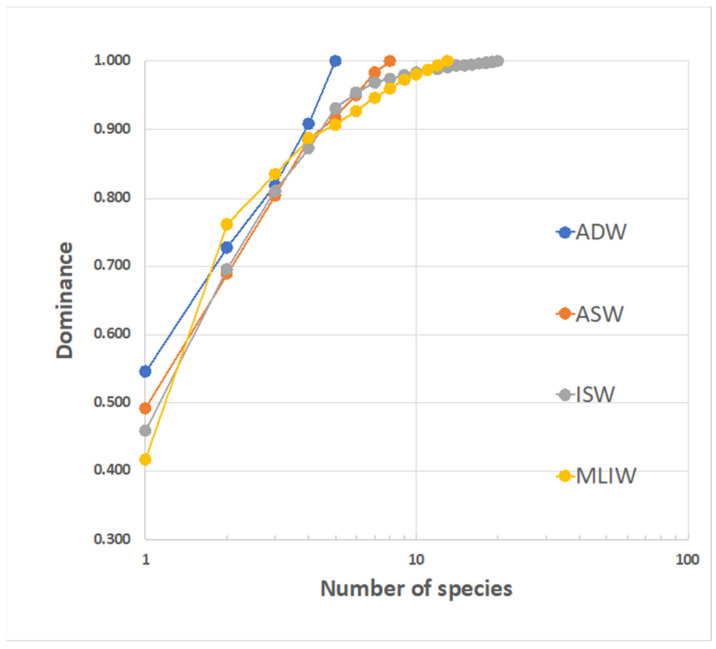
K-dominance curves for larvae assemblages corresponding to the different water masses identified in the area. The dominant species is *C. braueri* (from 42 to 55%). Species that are exclusive to a single water mass are: ASW: *Serranus* sp.; ISW: *L. crocodilus, B. glaciale, Lobianchia* sp., *T. thynnus* and unidentified species; MLIW: *A. hemigymnus, C. sloani*, *D. holti* and *L. pusillus*; ADW: *P. speciosa.*

**Table 1 biology-12-01449-t001:** Sampling details of COCONet-WP11 May 2013 cruise in the south Adriatic Sea.

Station	Local Date	Position		Local Time		Bottom Depth	Max Sampled Depth	No. of Samples
		Lat. N	Long. E	Start	End	(m)	(m)	
S1	9-May-2013	42°09.994′	15°39.966′	20:23	21:37	99	90	9
L41	10-May-2013	41°59.952′	16°59.872′	18:38	20:21	580	550	9
S3	10-May-2013	42°09.985′	16°38.059′	14:05	15:47	178	170	8
S10	11-May-2013	41°29.780′	18°22.453′	23:48	02:07	1123	1096	9
S7	11-May-2013	42°10.049′	18°32.310′	15:28	16:46	190	180	9
S8	12-May-2013	41°29.971′	18°50.127′	04:47	06:35	324	310	9
S16c	13-May-2013	40°53.072′	18°57.210′	11:48	13:09	317	300	8
S15	13-May-2013	41°02.365′	18°31.554′	05:36	07:38	939	900	8
S22	14-May-2013	40°05.222′	19°21.708′	18:03	20:25	965	900	9
S21	14-May-2013	40°05.008′	19°08.001′	22:17	00:28	972	900	8
S23	15-May-2013	39°40.001′	19°22.009′	18:01	20:29	1172	1100	9
S24	15-May-2013	39°40.004′	19°08.008′	22:14	00:23	1089	1000	9
S20	16-May-2013	40°05.012′	18°50.071′	14:50	16:41	738	700	9
S25	16-May-2013	39°39.917′	18°22.140′	04:26	05:59	261	210	8
S19	17-May-2013	40°26.801′	18°32.195′	10:19	11:42	127	100	7
S14	17-May-2013	41°02.305′	17°52.030′	18:02	19:51	699	600	9
S11	18-May-2013	41°29.991′	17°34.972′	07:10	09:25	1137	1060	9

**Table 2 biology-12-01449-t002:** Fish species identified in the study area: number of counted specimens (N), percentage contribution of each species, percentage frequency of occurrence in the sampling stations and total mean weighted abundance across all stations.

Family/Species	N	Relative Abundance	Frequency	Mean Abundance
		(%)	(%)	(ind. m^−2^)
Carangidae				
*Trachurus trachurus*	1	0.09	5.88	0.04
Cepolidae				
*Cepola rubescens*	6	0.53	23.53	0.41
Engraulidae				
*Engraulis encrasicolus*	268	23.59	64.71	27.58
Gonostomatidae				
*Cyclothone braueri*	518	45.60	94.12	45.40
*Cyclothone pygmaea*	71	6.25	64.71	6.28
*Cyclothone* sp.	6	0.53	11.76	0.37
Labridae				
*Coris julis*	4	0.35	11.76	0.40
Myctophidae				
*Benthosema glaciale*	2	0.18	5.88	0.21
*Diaphus holti*	1	0.09	5.88	0.14
*Ceratoscopelus maderensis*	115	10.12	70.59	10.16
*Hygophum benoiti*	5	0.44	17.65	0.34
*Hygophum* sp.	55	4.84	70.59	3.95
*Lampanictus pusillus*	1	0.09	5.88	0.14
*Lampanictus crocodilus*	3	0.26	11.76	0.30
*Lobianchia* sp.	1	0.09	5.88	0.11
*Myctophum punctatum*	18	1.58	35.29	1.47
Paralepitidae				
*Paralepis speciosa*	1	0.09	5.88	0.12
Photychthaidae				
*Vinciguerria attenuata*	33	2.90	29.41	1.98
*Vinciguerria* sp.	12	1.06	17.65	1.89
Pomacentridae				
*Chromis chromis*	2	0.18	11.76	0.22
Serranidae				
*Serranus* sp.	2	0.18	5.88	0.03
Sgombridae				
*Auxis rochei*	5	0.44	17.65	0.50
*Thunnus thynnus*	1	0.09	5.88	0.11
Sternoptychidae				
*Argyropelecus hemigymnus*	2	0.18	5.88	0.11
Stomiidae				
*Chauliodus sloani*	1	0.09	5.88	0.17
Unidentified specimens	2	0.18	11.76	0.16

**Table 3 biology-12-01449-t003:** Diversity descriptors for the fish larvae assemblages in the southern Adriatic Sea.

Water Mass	No. of Specimens	Species Richness	Margalef Index	Alpha Diversity	Whittaker’ Species Turnover
ASW	61	8	1.7	2.4	2.3
ISW	913	20	2.78	2.06	8.7
MLIW	151	13	2.39	1.75	6.4
ADW	11	5	1.66	1.2	3.2
*Overall*	1136	26	3.55	1.96	12.2

## Data Availability

The data presented in this study are available from Roberta Minutoli, rminutoli@unime.it.

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
