# Peer review of "Assemblage Structure of Ichthyoplankton Communities in the Southern Adriatic Sea (Eastern Mediterranean)"

_biology, 2023, doi:10.3390/biology12111449_

Round 1

Reviewer 1 Report

Comments and Suggestions for Authors

This paper is a good detailed description of the ichthyoplankton community within the area and is complete in the sense of type of analyses possible with the data. All aspects are covered and illustrated with good graphs. Except, the paper fails in delivering an appropriate discussion, which is weak in relation to structure and putting the results together in an integrative way. I have some suggestions for improving his, in my detailed comments. Most of those comments are small remarks, which are helping to finalize the manuscript. After those adaptations, especially discussion, the paper is worthwhile to be published.

To help in further improving the manuscript and to illustrate my general comments , I have some more detailed comments:

-        Line 43-44; “the” fits not in the sentences

-        Line 44: pelagic à used in the manuscript several times, but in intro you have the categories epipelagic, meso and bathypelagic. So is pelagic here epipelagic? Check consistency of this throughout the manuscript.

-        Line 48: “as” fits not in the sentence, correct

-        Line 100: food availability, specify between brackets what.

-        Line 122-123: COCONet project link, but any MPA or link with it is not mentioned in the manuscript? Is there or not relevant in this context? Give some explanation.

-        Is it right that at each station 10 nets were taken. All successful? Otherwise indicate in the table 1 how much there are taken/analysed.

-        Section 2.3. In results you have data on zooplankton, but is not mentioned here that this component is analysed or where is this coming from?

-        Line 151-152: Selection of station samples day or night is due to practical reason or any strategy? Explain.

-        Line 185: graphical position/area are five levels. Which, as geographically there are 3, so what is meant with area. Explain

-        Section 2.5: the diversity indices used in results are not listed here, so sentence 451-453 should move to this, alongside the other indices.

-        Section 3.2: Not explained in Mat&meth were this data come from

-        Sentence 256-257: re-write, not logic

-        Line 308-309: Highest values in which? Larvae or juveniles or all…

-        Line 331: Adult stage is mentioned here (also at other parts in result text), but data of adult abundances or .. not considered in manuscript, so where is this coming from? Clarify

-        Discussion is too chaotic, seems bullet points put in one text. Also the English writing is less good (e.g. lines 492-500 hard to follow and no logic word order) compared to the rest of the manuscript.

-        First part (line 457-479) is a general literature review, so use this as umbrella to pass in the result of this study or where this study contribute in those general statements.

-        In the discussion, it should be good to mention what the importance is of the species researches, from commercial and/or conservation purpose or importance in food web, ….

-        Line 508: “many spring coastal spawning fish”: What is this?

-        Line 514-516 is a result, not popping up there, so move it.

-        And last part on C braueri should be added to lines 484-487. And what in discussing the other 3 important species.

-        What is the explanation for the gradient from Italy to Albanian, is not clearly explained, discussed, which should be done.

-        So, please re-structure the discussion, for example to group the discussion in blocks of the horizontal, vertical and dominant species. And conclude with some recommendations on following this up, or usefulness to manage the fishery in the area or nature conservation.

Comments on the Quality of English Language

Some sentence or some parts (e.g. discussion) need to be checked, as they does not read correct.

Reviewer 2 Report

Comments and Suggestions for Authors

Studies based on fish early life stages can provide information on spawning grounds and nursery areas, with implications on stock biomass fluctuations induced by recruitment variability. This study describes the composition, abundance, spatial distribution, diel vertical patterns of ichthyoplankton in the Southern Adriatic Sea. The results are basic, their significance should be improved. 

Comments on the Quality of English Language

Extensive editing of English language required.

Reviewer 3 Report

Comments and Suggestions for Authors

The manuscript presents the results of a single sampling at 17 stations. The collection of material was carried out for only 10 days in May. The authors do not consider the interaction of fish larvae with each other. Also, the manuscript does not consider the influence of sea currents and wind on the placement of fish larvae. In essence, the results of studying the diversity of fish larvae (ichthyoplankton) at the end of spring are presented. Therefore, the title of the manuscript does not reflect the content of the work.

Sampling was carried out at different stations at different times of the day, but this does not give grounds to judge daily changes in the distribution of fish larvae. For a correct study of diurnal changes in the vertical distribution of fish larvae, all samples had to be taken at each station during the day, and then the results obtained should be compared with the WMD model (lines 166–173). Therefore, the authors did not solve the task of studying the daily distribution of fish larvae (line 98).

The journal Biology is intended for a wide audience of biologists, so the authors need to describe in more detail how the sampling of ichthyoplankton was carried out at different depths (layers).

In the "Materials and Methods" section, it is also necessary to provide a link to the methods for calculating diversity indicators, which are given in Table 3.

It is not clear why the authors provided generalized data on zooplankton (Section 3.2, lines 230-236?

Figure 1 indicates 18 sampling stations, but the authors discuss only 17 (line 124). In the same figure, it is necessary to carry out a continuous and uniform numbering of stations. Why is one of the stations marked "L41", and all the others "S ..."?

Why are there no stations 2, 5, 6, 9, 12, 13, 17, 18?

All drawings must be legible.

There are unnecessary self-citations - 7 sources from the list of references belong to the authors of this manuscript.

Line 258. Typo in name should be scombrid instead of sgombrid

Perhaps, for a better understanding of the manuscript, one should compare the species diversity using the Sörensen or Bray-Curtis coefficient

Reviewer 4 Report

Comments and Suggestions for Authors

The manuscript focus on the assemblage structure of ichthyoplankton communities. This study is good work, but some questions need to be revised.

General comments:

1.The title need to be revised. Specially, ... in relation to oceanographic conditions”

2.Line 38-40 and 44-47: The abstract need to be revised. The abstract should present the background of the study, the main research and results, and the conclusions. It is inappropriate for a lot of numbers to appear in a paragraph.

Special comments:

3.Line 95-100: Consider combining them into one paragraph to highlight the motivation and content of the study.

4.Line 405-416: The p-value needs to be italicized.

5.Line 480-486: Add relevant discussion to enhance the persuasiveness of the results.

Round 2

Reviewer 3 Report

Comments and Suggestions for Authors

Line 142. Probably, it would better to write " The opening and closing of each net were ...."